# Isolation and Pathogenicity Analysis of a G5P[23] Porcine Rotavirus Strain

**DOI:** 10.3390/v16010021

**Published:** 2023-12-22

**Authors:** Liguo Gao, Hanqin Shen, Sucan Zhao, Sheng Chen, Puduo Zhu, Wencheng Lin, Feng Chen

**Affiliations:** 1College of Animal Science, South China Agricultural University, Guangzhou 510642, China; gaoliguo@stu.scau.edu.cn (L.G.); zhaosc@stu.scau.edu.cn (S.Z.); chens@stu.scau.edu.cn (S.C.); zpd@stu.scau.edu.cn (P.Z.); wenchenglin@scau.edu.cn (W.L.); 2Wen’s Food Group, Yunfu 527300, China; hqshen@wens.com.cn; 3Guangdong Jingjie Inspection and Testing Co., Ltd., Yunfu 527300, China

**Keywords:** porcine rotavirus, G5, virus isolation, pathogenicity

## Abstract

(1) Background: Group A rotaviruses (RVAs) are the primary cause of severe intestinal diseases in piglets. Porcine rotaviruses (PoRVs) are widely prevalent in Chinese farms, resulting in significant economic losses to the livestock industry. However, isolation of PoRVs is challenging, and their pathogenicity in piglets is not well understood. (2) Methods: We conducted clinical testing on a farm in Jiangsu Province, China, and isolated PoRV by continuously passaging on MA104 cells. Subsequently, the pathogenicity of the isolated strain in piglets was investigated. The piglets of the PoRV-infection group were orally inoculated with 1 mL of 1.0 × 10^6^ TCID50 PoRV, whereas those of the mock-infection group were fed with an equivalent amount of DMEM. (3) Results: A G5P[23] genotype PoRV strain was successfully isolated from one of the positive samples and named RVA/Pig/China/JS/2023/G5P[23](JS). The genomic constellation of this strain was G5-P[23]-I5-R1-C1-M1-A8-N1-T1-E1-H1. Sequence analysis revealed that the genes *VP3*, *VP7*, *NSP2*, and *NSP4* of the JS strain were closely related to human RVAs, whereas the remaining gene segments were closely related to porcine RVAs, indicating a reassortment between porcine and human strains. Furthermore, infection of 15-day-old piglets with the JS strain resulted in a diarrheal rate of 100% (8 of 8) and a mortality rate of 37.5% (3 of 8). (4) Conclusions: The isolated G5P[23] genotype rotavirus strain, which exhibited strong pathogenicity in piglets, may have resulted from recombination between porcine and human strains. It may serve as a potential candidate strain for developing vaccines, and its immunogenicity can be tested in future studies.

## 1. Introduction

Rotaviruses (RVs) are the most common cause of viral gastroenteritis in infants and animals. RVs are primarily transmitted through the fecal–oral route and infected individuals typically present with symptoms such as diarrhea, vomiting, and decreased appetite. The infection can spread between humans and animals, posing a serious threat to both human health and animal husbandry [1,2,3,4]. In 1969, bovine RV was first isolated and cultured in cells and was confirmed to be the primary cause of diarrhea in cattle [5]. The first human RV was discovered in 1973 [6]. RVs were subsequently identified in pigs and poultries [7,8]. Cultivating RVs in cell cultures was quite challenging until 1984. Subsequently, RVs were successfully cultured in African Green Monkey kidney cells (MA104) after treatment with trypsin, which accelerated the processes of RV isolation and cultivation [9]. Furthermore, subsequent studies have reported that RVs could replicate in cells such as Vero cells [10,11].

Porcine rotavirus infection mainly causes diarrhea in suckling piglets, manifesting as watery diarrhea, weight loss, and dehydration [12]. Porcine rotavirus infection occurs in piglets aged 1 to 4 weeks, and PoRV is co-infected with other viral pathogens such as Porcine Epidemic Diarrhea Virus (PEDV) and Transmissible Gastroenteritis Virus (TGEV), as well as bacterial pathogens such as *Escherichia coli*, causing significant losses to the pig farming industry [13,14]. Porcine rotaviruses (PoRVs) infection can damage the small intestines in pigs, leading to villous atrophy and reduced digestive absorption capacity [15]. The most important mechanism for RVA to cause secretory diarrhea is that NSP4 serves as an enterotoxin that increases chloride secretion through a calcium-dependent mechanism, and it also activates the enteric nervous system and blocks the intestinal sodium/glucose cotransporter [16,17].

The genus rotavirus belongs to the family Reoviridae and order Reovirales. Their genome consists of 11 double-stranded RNA segments, encoding 11 proteins, including 6 structural proteins (virion protein [*VP*]*1*–*VP4*, *VP6*, and *VP7*) and 5 nonstructural proteins (*NSP1–NSP5/6*) [15]. RVs are classified into 10 subgroups (A–J) based on the *VP6* antigen [18]. The outer shell proteins *VP7* and *VP4* induce neutralizing antibodies and form the basis for the G and P dual classification system [19]. The most common groups infecting Porcine are Group A RV (RVA), Group B RV (RVB), and Group C RV (RVC), with RVA having the highest prevalence and causing the most significant harm [15]. For RVA strains with high genetic diversity, the dual (G/P) typing system was expanded in 2008 to a complete genome sequence-based classification system. A nomenclature for the comparison of complete rotavirus genomes was considered in which the notations Gx-P[x]-Ix-Rx-Cx-Mx-Ax-Nx-Tx-Ex-Hx are used for the *VP7-VP4-VP6-VP1-VP2-VP3-NSP1-NSP2-NSP3-NSP4-NSP5/6* encoding genes, respectively. Subsequently, the Rotavirus Classification Working Group (RCWG) was established to develop classification guidelines and maintain the proposed classification system of RVs [20]. To date, 42 “G” and 58 “P” genotypes have been reported in the RCWG database (https://rega.kuleuven.be/cev/viralmetagenomics/virus-classification/rcwg, accessed on 1 October 2023). RVA strains with G3, G5, G9, and G11 genotypes in combination with P[9], P[6], P[13], P[23], P[27], and P[28] genotypes are considered the most common worldwide [21].

In this study, a G5P[23] genotype PoRV strain was isolated from the diarrheal samples of piglets on a farm in Jiangsu Province, China, and named RVA/Pig/China/JS/2023/G5P[2](JS). The whole genome of the JS strain was sequenced, and its genetic evolution was determined. The JS strain was also used to infect 15-day-old piglets to investigate its pathogenicity.

## 2. Materials and Methods

### 2.1. Clinical Samples, Cells, and Antibodies

On 6 January 2023, 28 diarrheal samples were collected from piglets in a pig farm infected by PoRV in Jiangsu Province, China. The virus isolated from these samples was identified as PoRV through Reverse Transcription-Polymerase Chain Reaction (RT–PCR) and Quantitative real-time polymerase chain reaction (qRT–PCR) using *VP6* (Appendix A) [22,23,24]. Samples of the intestinal contents were homogenized in serum-free Dulbecco’s modified Eagle medium [DMEM] (Invitrogen, Carlsbad, CA, USA) containing 1% penicillin–streptomycin [10,000 units/mL penicillin and 10,000 µg/mL streptomycin] (Gibco™, New York, MT, USA) and 0.3% trypsin phosphate broth (Sigma-Aldrich, St. Louis, MO, USA). The samples were then centrifuged for 30 min at 3500 rpm and 4 °C. The supernatant was filtered through a 0.22 µm pore filter (Merck Millipore, Darmstadt, Germany) to remove the bacteria and was stored at −80 °C until use as an inoculum for virus isolation. MA104 cells were cultured in DMEM supplemented with 10% heat-inactivated fetal bovine serum [FBS] (Invitrogen, Carlsbad, CA, USA) and 1% antibiotics [10,000 units/mL penicillin, 10,000 µg/mL streptomycin, and 25 µg/mL Fungizone^®^] (Gibco™, New York, MT, USA). These cells were maintained at 37 °C in a humidified incubator under 5% CO_2_. Mouse monoclonal antibodies (McAbs) against PoRV *VP6* protein (Zoonogen^®^, Beijing, China) were purchased and stored in the laboratory.

### 2.2. Virus Isolation Assay

PoRV isolation was performed using MA104 cells via a previously described method with modifications [25]. Specifically, prior to inoculation, cells were rinsed thrice with sterile phosphate-buffered saline (PBS; pH 7.2) to remove FBS completely. Simultaneously, the stored inoculum was briefly vortexed and used for cultivation. In total, 300 µL of the inoculum was added to 6-well plates. After incubation at 37 °C for 1 h, the inoculum was removed, and the maintenance medium containing trypsin at a final concentration of 4 µg/mL was added. The inoculated cells were maintained at 37 °C in a humidified incubator under 5% CO_2_, and the CPEs were monitored daily. When CPEs were observed in 90% of the cells, the flask was subjected to three rounds of freezing and thawing. The cells and the supernatant were mixed with a pipette, aliquoted, and stored at −80 °C. The harvested cell culture was used as the seed stock for the next generation. For successive passages, the scale of the culture was gradually increased until the PoRV strains were properly propagated and continuously passaged using T-25 flasks. Subsequently, the virus was purified using the limited-dilution method.

### 2.3. Titration and Growth Curve Determination

The growth curve of the 10th generation of the virus was constructed. Initially, the virus was inoculated onto a 96-well plate at a multiplicity of infection (MOI) of 0.01. Cell culture supernatants were collected at 6, 12, 18, 24, 30, 36, 42, and 48 hpi. Subsequently, based on the Reed–Muench method [26], the viral titers of these samples were determined using the TCID50 assay. After washing thrice with PBS, 100 µL of the experimental supernatant was mixed with 900 µL of DMEM to obtain serial dilutions ranging from 10^−1^ to 10^−10^. Each dilution was added to a monolayer of MA104 cells in 10 vertical wells of a 96-well plate. The plate was then placed in a cell culture incubator at 37 °C for 5 days, and the viral titers were determined by observing the CPEs under a microscope. Finally, the growth curves for each virus were constructed based on the viral titers at different time points post-infection.

### 2.4. Immunofluorescence Assay (IFA)

MA104 cells cultured in 6-well plates were either mock-infected or infected with PoRV at a MOI of 0.01. At 0, 12, and 24 hpi, the cells were fixed with 4% paraformaldehyde at 4 °C for 30 min, followed by permeabilization with 0.25% Triton X-100 (Solarbio, Beijing, China) for 10 min at room temperature (RT). Blocking was performed using 5% bovine serum albumin (Solarbio, Beijing, China) in PBS at RT for 1 h. McAbs against the PoRV *VP6* protein and Alexa Fluor^®^ 488-conjugated goat anti-mouse IgG (Abcam, Cambridge, UK) were used as primary and secondary antibodies, respectively. Nuclei were stained with 4′,6-diamidino-2-phenylindole (DAPI) (VectorLabs, Newark, CA, USA) for 20 min at RT. After washing with PBS, the stained cells were observed under a fluorescence microscope (Olympus, Tokyo, Japan).

### 2.5. Transmission Electron Microscopy (TEM)

To visualize PoRV particles, MA104 cells were harvested when CPEs were observed in 90% of the cells. The cell culture was subjected to three freeze–thaw cycles, followed by centrifugation for 30 min at 4000× *g* and 4 °C. The supernatant was filtered through a 0.22 mm filter to remove cell debris and was then mixed overnight with polyethylene glycol 8000 [PEG-8000] (Solarbio, BeiJing, China) at a final concentration of 10%. The mixture was subsequently ultracentrifuged for 2 h at 118,000× *g* and 4 °C to obtain the PoRV particles. These particles were then resuspended in Tris-buffered saline and negatively stained with 2% phosphotungstic acid. The viruses within the infected MA104 cells were examined using a transmission electron microscope (TEM) (Leica, Wetzlarm, Germany), and further fixation and imaging were performed based on the methods described in previous studies.

### 2.6. Sequence Analysis

The complete genome of the RVA strain JS was sequenced using the MiSeq high-throughput sequencing platform (Illumina, San Diego, CA, USA). The total RNA from the cell culture supernatant was extracted using TRIzol reagent (Invitrogen, Carlsbad, CA, USA) following the manufacturer’s instructions. rRNA was removed using RiBo-Zero Magnetic Gold kit (Epicenter Biotechnologies, Madison, WI, USA), and the remaining RNA was sequenced using NEBNext^®^ Ultra™ Directional RNA Library Prep Kit (New England Biolabs, Ipswich, MA, USA) and the Illumina MiSeq platform (GENEWIZ, Guangzhou, China).

The obtained raw reads were trimmed using Trimmomatic v0.39 software [16] and then aligned with the Sscrofa 11.1 reference genome using Bowtie2 v2.4.1 software [17] to eliminate the reads corresponding to the sequences of the pig genome. The remaining reads were reassembled into contigs using MEGAHIT v1.2.9 software [18]. Viral sequences were identified using BLASTn and BLASTx searches against a custom-built virus nucleotide reference database from GenBank or UniProt virus classification database.

The genotypes of the isolated strains were determined using the automatic genotyping tool provided by ViPR (https://www.viprbrc.org/, accessed on 6 October 2023), with a nucleotide truncation threshold of 80% [27]. To further explore the genetic origin of JS, phylogenetic trees were constructed using the neighbor-joining method using MEGA11 software with the default settings (1000 bootstrap replicates) based on the PoRV *VP4* and *VP7* gene sequences sequenced in this study and reference sequences collected in GenBank.

### 2.7. Recombination Analysis

The sequences involved in recombination were analyzed using RDP4 software version 4 [28]. DP4: Detection and analysis of recombination patterns in virus genomes. Virus Evol. 1, vev003 [28]. The RDP, GENECONV, Chimaera, MaxChi, Bootscan, SiScan, and 3Seq methods were employed with their default parameters.

### 2.8. Animal Experiments

Sixteen conventional piglets, 15 days old, were purchased from a commercial pig farm with no history of PoRV outbreaks or vaccination. Additionally, various virus tests, including qRT-PCR and enzyme-linked immunosorbent assay (ELISA), yielded negative results for these piglets. The tested viruses included PoRV, PEDV, TGEV, Porcine Reproductive and Respiratory Syndrome Virus (PRRSV), Porcine Circovirus (PCV), Classical Swine Fever Virus (CSFV), and Pseudorabies Virus (PRV). All animal experiments were approved by the Ethics Committee of the South China Agricultural University, Guangzhou, China (approval ID: SYXK-2019-0136). The 18 piglets were randomly divided into 2 groups, with 8 piglets in each group: PoRV-infection and mock-infection groups. The piglets of the PoRV-infection group were orally inoculated with 1 mL of 1.0 × 10^6^ TCID50 PoRV, whereas those of the mock-infection group were fed with an equivalent amount of DMEM. The animals were manually fed with a milk replacer every 4 h during the experiment. In piglets showing signs of anorexia, 60–100 mL of milk replacer was administered by gavage every 4 h. Clinical symptoms such as diarrhea, vomiting, anorexia, and depression were recorded daily. Fecal consistency was assessed daily using a scoring system based on solid (0 points), paste-like (1 point), semiliquid (2 points, mild diarrhea), and liquid (3 points, severe diarrhea) standards. Three piglets from the virus-infected group and three piglets from the mock group were euthanized at 9 d post-infection (dpi).

Histopathological and immunohistochemical examinations were performed on the small intestines. Intestinal tissue samples from a randomly selected piglet were examined and collected 9 dpi. After fixation in 4% formaldehyde at RT for 48 h, the processed tissue samples were embedded in paraffin, sectioned using a microtome (Leica, Germany), deparaffinized with xylene, and washed with reducing concentrations of ethanol. Subsequently, conventional Hematoxylin and Eosin (BaSo, Zhuhai, China) [H&E] staining was performed for HE, and PoRV-VP6-specific monoclonal antibody was used IHC [29].

Rectal swabs were collected daily for the qRT–PCR detection. The animals were euthanized 9 dpi, and the tissues of the lungs, duodenum, jejunum, ileum, etc., were collected for HE, IHC, and qRT–PCR examination.

### 2.9. Statistical Analysis

All statistical analyses were performed using the GraphPad Prism software version 8.0 (graphpad.com). The statistical significance of the differences between the experimental groups was ascertained using the Student’s *t*-test. The differences were considered statistically significant at *p* < 0.05.

## 3. Results

### 3.1. Virus Isolation

The collected piglet diarrhea samples were tested using RT-PCR and qRT-PCR, and all 28 diarrhea samples showed positive results for porcine rotavirus. Inoculate these 28 positive samples into MA104 cells for three blind passages to isolate the virus. Only one strain was successfully isolated from positive samples, while the remaining samples could not be continuously passaged on MA104 cells.

The PoRV-positive samples at a 10-fold dilution were inoculated into MA104 cells. Visible cytopathic effects (CPEs) were observed in the second generation, 24 h post-infection [hpi]. Complete CPE was observed at 48 hpi (Figure 1a). Compared with mock-inoculated cells, the inoculated cells were initially characterized by syncytium and vacuole formation, followed by elongation, detachment, and cessation of growth (Figure 1a). The isolate was named RVA/Pig/China/JS/2023/G5P[23](JS).

Furthermore, the isolated strain was identified via IFA using PoRV-VP6-specific monoclonal antibodies. From 12 hpi, pronounced green signals were observed in the infected MA104 cells but not in the uninfected cells. These signals tended to intensify markedly with time (0, 12, and 24 hpi) (Figure 1b).

TEM was also conducted to identify PoRV particles obtained from the MA104 cell culture medium and MA104 cells infected with the JS strain. The typical RV particles were observed in the cell culture medium (Figure 1c). The viral particles appeared wheel-shaped, with a diameter of 70–100 nm, exhibiting characteristic surface projections unique to RVs. TEM confirmed the successful reproduction of PoRV in MA104 cells. The growth curves of the JS strain at different time points post-infection were plotted based on the 50% tissue culture infective dose (TCID50). The results indicated that the viral replication titer peaked at ~1 × 10^6^/mL TCID50, and the titer enhanced rapidly from 6 to 36 hpi (Figure 1d).

### 3.2. Sequence Analysis

According to the latest classification and naming system established by the RCWG, the nucleotide identity cutoff values for 11 gene segments of Rotavirus Group A are as follows: 80% (G), 80% (P), 85% (I), 83% (R), 84% (C), 81% (M), 79% (A), 85% (N), 85% (T), 85% (E), and 91% (H) [30]. The genotype of JS was identified as G5-[P23]-I5-R1-C1-M1-A8-N1-T1-E1-H1 (Appendix A). The sequences of all 11 gene segments were uploaded to GenBank with accession numbers OR644644 to OR644654.

Homology and phylogenetic analyses were conducted between the closely related reference RVA strains of the same genotype. Several gene segments were highly homologous with RV strains of porcine origin, including *VP1*, *VP2*, *VP4*, *VP6*, *NSP1*, *NSP3*, and *NSP5*. Additionally, some segments such as *VP3*, *VP7*, *NSP2*, and *NSP4* exhibited an even higher homology with RV strains of human origin, suggesting that the JS-G5P[23] strain was likely a recombinant human–porcine virus (Appendix A).

Sequence analysis of *VP7* revealed that it comprised 981 nucleotides (nts), encoding 326 amino acids (aa). A phylogenetic tree was constructed using the full-length *VP7* sequence (981 nts) and selected G genotype sequences obtained from GenBank. The *VP7* of the JS strain is grouped within the G5 genotype (glycolated), primarily consisting of strains of porcine origin. However, the segment from the JS strain formed a smaller clade within this genotype, containing sequences derived from both pig and human sources. The most closely related sequence to the JS strain fragment was identified to be derived from a human RVA strain, RVA/Pig-wt/THA/CMP-001-12/2012/G5P[13] (KT727252.1), which was isolated from a patient in Thailand in 2014 (Figure 2a) [31]. The nucleotide identity rates between the selected G5 genotypes in GenBank range from 86.6% to 91.4%, and the amino acid identity rates range from 92.0% to 95.6%. Furthermore, using ViPR automatic genotyping tool, we revealed that RVA/Pig/China/JS/2023/G5P[23] was closely related to multiple RVA G5 strains [27]; this sequence belongs to the G5 genotype (glycolated), with the most similar query sequence being RVA/Human-tc/BRA/IAL28/1992/G5P[8] (EF672588) (Appendix A) [32].

Sequence analysis of *VP4* revealed that it comprised 2331 nts, encoding 776 aa. A phylogenetic tree was constructed using the full-length *VP4* sequence from the JS strain and selected P genotype sequences obtained from GenBank. JS was most closely related to a porcine-origin RVA strain, RVA/Pig/CHN/SX-2021/P23 (OP650547.1), isolated from China (Figure 2b). The nucleotide identity rates between the selected [P13] genotypes in GenBank range from 80.0% to 89.6%, and the amino acid identity rates range from 72.0% to 83.1%. Additionally, the ViPR automatic genotyping tool revealed that RVA/Pig/China/JS/2023/G5P[23] was closely related to RVA P[23] strains [21]; this sequence belongs to the P[23] genotype, with the highest similarity to the PoRV strain P23-RVA/Pig-tc/VEN/A34/1985/G5P[23] (AY174094) (Appendix A) [27,33].

### 3.3. Recombination Analysis

To analyze the association between *NSP2* from JS and the existing isolates further, a genetic analysis was conducted between them using RDP4 software [28]. The sequence of *NSP2* from JS is indicated in Figure 3. Crossover points for a potential recombination zone were located at nucleotides 1–494 and 939–994 of *NSP2* (Figure 3), but no recombination was detected in other genes encoding viral proteins. The major parental strain for the recombination was RVA/Human-tc/KOR/CAU14-1-262/2014/G3P[9] (KR262156.1), and the minor strain was RVA/Dog-wt/GER/88977/2013/G8P1/(KJ940158.1), indicating that C1 of JS was the gene recombination product of C3 and C2 RVAs [34,35].

### 3.4. Clinical Signs and Histological Changes

The piglets of the JS strain infection group showed diarrhea symptoms, manifested as dark green, watery feces (Figure 4a). One piglet succumbed on day 3 post-infection and two piglets on day 6. On autopsy, the lungs showed congestion and swelling, the intestinal wall became thin and transparent, and the intestines bulged and were filled with yellow water-like liquid (Figure 4a). On the contrary, in the simulated infection group, except for a few piglets whose feces briefly appeared pasty, none experienced diarrhea, and none succumbed at the end of the experimental cycle. No pathological changes were found in the lungs and intestinal tissues of these piglets during autopsy (Figure 4a).

The HE results indicate that the lung exhibited thickening of the alveolar walls, suggesting interstitial pneumonia (as shown by the green arrow in Figure 4b). The villi lamina propia shows an increased number of inflammatory cells, predominantly lymphocytes (as shown by the black arrow in Figure 4b). The intestinal tissue pathology of piglets in the control group was normal (Figure 4b). In addition, IHC examination revealed that in specific segments of the small intestine with villous atrophy, cytoplasmic staining of the PoRV antigen was dominant, with the highest antigen content in the jejunum (Figure 4c). No PoRV antigen was detected in the small intestines of piglets in the negative control group (Figure 4c).

### 3.5. Viral Load in Stool Samples and Tissues

One piglet developed diarrhea on day 2 post-oral inoculation with the JS strain (Figure 5a). By day 3, all eight piglets exhibited diarrhea; one of them succumbed to the infection on day 3 and two on day 6 (Figure 5b). From day 7 onward, most piglets began to eat, which continued until day 9, when the symptoms of diarrhea began to subside. The piglets infected with JS had a diarrheal rate of 100% (8 of 8; Figure 5a) and a mortality rate of 37.5% (3 of 8; Figure 5b). In contrast, the control group, apart from a brief period of loose stools in a few piglets, did not show any significant signs of diarrhea, and no deaths occurred (Figure 5a,b).

qRT–PCR results showed that the viral RNA was detectable in the feces of infected piglets as early as 24 h post-infection (hpi). Viral shedding in rectal swabs of piglets continued to increase from day 1 to day 5 post-infection, reaching its peak on day 5 (Figure 5c). From day 7, viral shedding began to decline and decreased until the end of the experiment. Additionally, the viral load in the lungs was significantly lower than in various parts of the small intestine (*p* < 0.05), with the highest viral load detected in the ileum (Figure 5d). No viral shedding was detected in the feces of the control group, and no viral load was identified in the intestinal tissues (Figure 5c,d).

## 4. Discussion

Approximately 50 years ago, RVs were considered the primary cause of diarrhea in infants and young animals [6]. According to the World Health Organization, RVs cause ~450,000 deaths annually, with 90% of them occurring in developing countries in Asia and Africa [4,36,37]. PoRVs are prevalent worldwide and result in significant economic losses to the swine industry [15]. Isolation and cell-based cultivation of PoRV can help understand their pathogenicity in piglets [38]. In this study, PoRV-positive samples from a pig farm, suffering from diarrhea in suckling piglets, were treated with 10 µg/mL trypsin for 1 h, and the optimal conditions for the isolation of PoRV were established by adding trypsin to a final concentration of 4 µg/mL in the maintenance medium. A PoRV strain was successfully isolated under these conditions and named JS. After three-blind-passaging in MA104 cells, stable CPEs, including cell elongation and detachment, were observed (Figure 1a), similar to previous studies [12,25]. IFA confirmed the replication of the 10th generation of the JS strain in MA104 cells (Figure 1b). TEM revealed typical RV particles of 70–100 nm in diameter in the cell culture medium (Figure 1c). The viral titer at the 10th passage was ~1 × 10^6^/mL TCID50 (Figure 1d).

The complete genome sequence of JS was obtained using high-throughput sequencing. Analysis of genetic evolution revealed that the JS strain was a G5P[23] genotype RV. The genotype G5 PoRV is more common in pigs than in horses, cows, or other animals. It was identified in children in Brazil in the 1980s [39,40]. Although the G5 genotype RV has been reported in multiple countries, research on its pathogenicity is limited [19]. ViPR-based analysis revealed that the genotype of JS strain is G5-P[23]-I5-R1-C1-M1-A8-N1-T1-E1-H1. The structural proteins *VP3* and *VP7*, along with the nonstructural proteins *NSP2* and *NSP4* of the RVA/Pig/China/JS/2023/G5P[23], were closely related to human RVAs. However, the structural proteins *VP1*, *VP2, VP4*, and *VP6*, along with the nonstructural proteins *NSP1*, *NSP3*, and *NSP5*, were closely related to porcine RVAs. This suggests that the JS strain is a recombinant strain of human and porcine RVAs (Appendix A). The *NSP2* of the JS strain may be a product of recombination between human and dog strains (Figure 3).

A previous study reported severe watery diarrhea in 3-day-old piglets within 24 h of infection with the HN03 (G9P[23]) strain and recovery after 72 h [12]. Similarly, diarrhea was observed in 4-day-old piglets at 16–24 h after infection with the JS-01-20149 (G9P[7]) strain, followed by skin redness and death [41]. In a previous study, diarrhea was reported in piglets infected with PRG942 (G9P[23]) and PRG9121 (G9P[7]) strains at 1–8 dpi [42]. Furthermore, Miao et al. reported diarrhea in 1-day-old piglets infected with the CN127 strain after 6–24 h, which continued until euthanization at 48 h, but none of them died due to the infection [38].

In this study, the pathogenicity of the JS strain was investigated by infecting 15-day-old piglets. Diarrhea and dark green, foamy feces were observed on day 2 post-infection. On day 3, one piglet succumbed to the infection (Figure 4a). A postmortem examination of the piglets in the JS-infected group showed transparent, distended intestines filled with yellowish fluid and the accumulation of gastric and intestinal gases. RVA infections are not restricted to the intestines and further induce the formation of extraintestinal lesions in humans and animals [43]. After dissection, the lungs of infected piglets also showed significant lesions, manifested as local congestion (Figure 4a). The HE results indicate that the lung exhibited thickening of the alveolar walls, suggesting interstitial pneumonia. The villi lamina propia shows an increased number of inflammatory cells, predominantly lymphocytes (Figure 4b). The intestinal tissue pathology of piglets in the control group was normal (Figure 4b). In addition, IHC examination revealed that in specific segments of the small intestine with villous atrophy, cytoplasmic staining of the PoRV antigen was dominant, with the highest antigen content in the jejunum (Figure 4c). No PoRV antigen was detected in the small intestines of piglets in the negative control group (Figure 4c).

At the end of the experimental period, the diarrheal and mortality rates of infected piglets were 100% (8 of 8) and 38.5% (3 of 8), respectively. In contrast, piglets in the control group did not exhibit any apparent symptoms of diarrhea, and no deaths occurred (Figure 5a,b). TaqMan qRT–PCR detected viral RNA shedding in the feces of infected piglets as early as 24 hpi. Viral shedding in the rectal swabs of piglets continued to increase from day 1 to day 5 post-infection but decreased from day 6 onward (Figure 5c). The viral load in the lungs was significantly lower than in various parts of the small intestine, such as the duodenum, jejunum, and ileum (*p* < 0.05). The highest viral load was detected in the ileum, consistent with the IHC results (Figure 5d). Throughout the entire experiment, no viral shedding was detected in the feces, and no viral load was identified in the intestinal tissues of the control group animals (Figure 5c,d).

There are differences between using artificial milk feeding instead of breastfeeding during the experimental period and production in this study. The study is based on samples collected from a specific farm in Jiangsu Province, China. The findings may not be representative of PoRVs in other regions or countries. In clinical pathology, the infection of porcine rotavirus is more complex, and it occurs in different age groups of pigs [15]. In addition, PoRV is co-infected with other viral pathogens, such as PEDV and TGEV, as well as bacterial pathogens, such as *Escherichia coli* [13]. These phenomena have brought new challenges to this research study.

## 5. Conclusions

In summary, we successfully isolated and identified a strain of G5P[23] genotype PoRV. This strain exhibited excellent adaptability to MA104 cells and demonstrated high virulence in piglets. These findings hold significant importance for understanding the characteristics of PoRV in China and developing novel and effective PoRV vaccines.

## Figures and Tables

**Figure 1 viruses-16-00021-f001:**
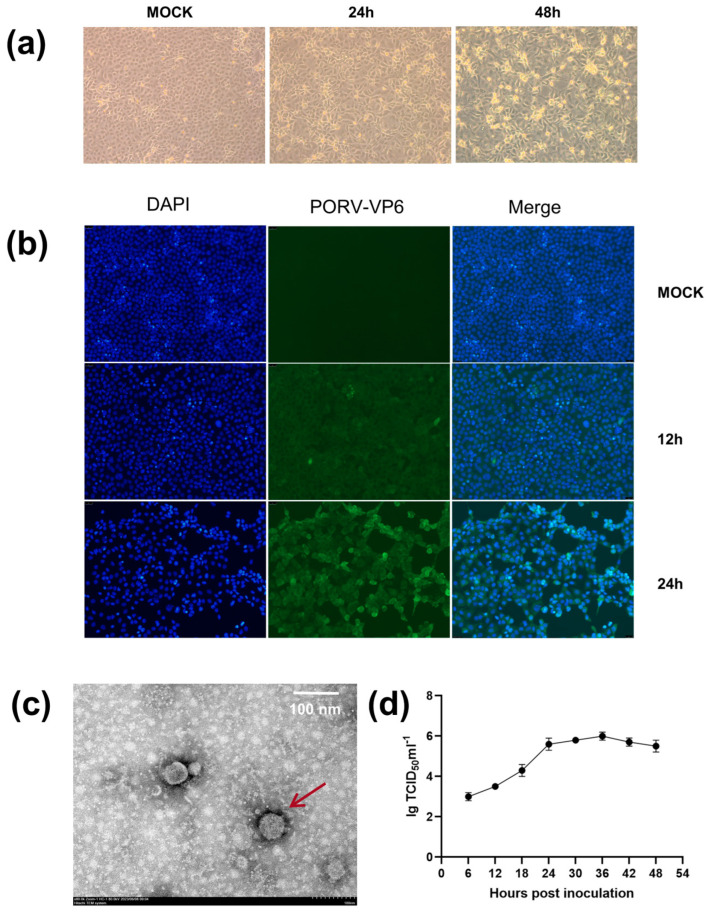
(**a**) Observation of CPEs in MA104 cells infected with the PoRV JS strain at 24 and 48 h post-infection (hpi). (**b**) Immunofluorescence staining (green) of PoRV *VP6* in MA104 cells infected with the JS strain at 12 and 24 hpi. (**c**) Electron micrograph of the PoRV particles detected in the culture medium of MA104 cells infected with the JS strain, as indicated by arrows. Scale bar = 100 nm. (**d**) Growth curve of the JS strain at the 12th passage of MA104 cells. Multiplicity of infection (MOI) = 0.01.

**Figure 2 viruses-16-00021-f002:**
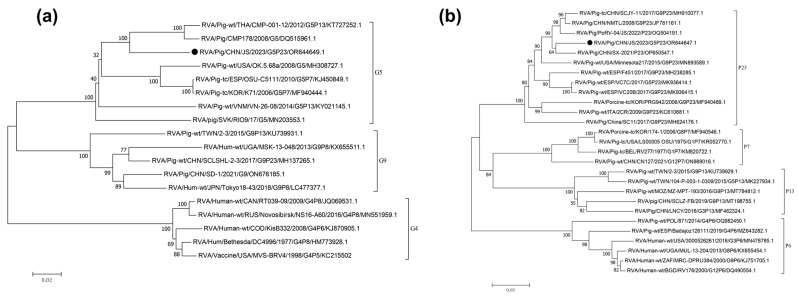
(**a**) Phylogenetic tree constructed using the segments of *VP7*; G = glycolated. (**b**) Phylogenetic tree constructed using the segments of *VP4*; P = protease-sensitive. ● Display JS strain sequence.

**Figure 3 viruses-16-00021-f003:**
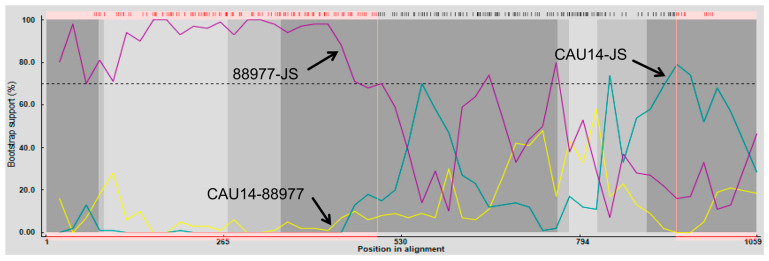
Analysis of the recombination of *NSP2* from RVA/Pig/China/JS/2023/G5P[23] and other PoRV strains such as RVA/Human-tc/KOR/CAU14-1-262/2014/G3P9/(KR262156.1) and RVA/Dog-wt/GER/88977/2013/G8P1/(KJ940158.1). JS represents RVA/Pig/China/JS/2022/G3P[7] (OR232953), CAU14 represents RVA/Human-tc/KOR/CAU14-1-262/2014/G3P[9] (KR262156.1), and 88977. represents RVA/Dog-wt/GER/88977/2013/G8P1/(KJ940158.1). Recombination crossover points are located at nucleotides 1–494 and 939–994.

**Figure 4 viruses-16-00021-f004:**
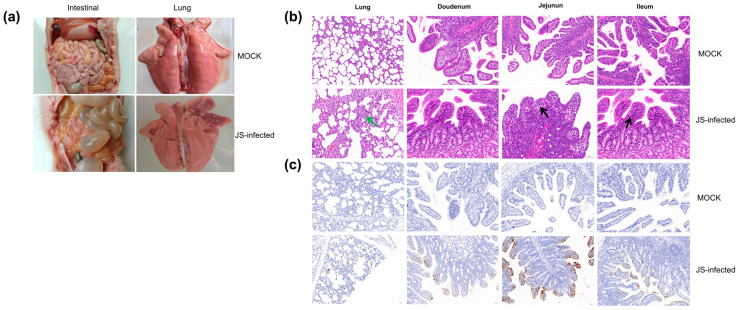
(**a**) Intestinal and lung sections of the piglets in the mock-infected and JS-infected groups. (**b**) Histopathological sections of the piglets in both groups. (Green indicates thickening of alveolar walls, while black indicates an increase in inflammatory cells) (**c**) Immunohistochemical staining of the tissues collected from the piglets in both groups.

**Figure 5 viruses-16-00021-f005:**
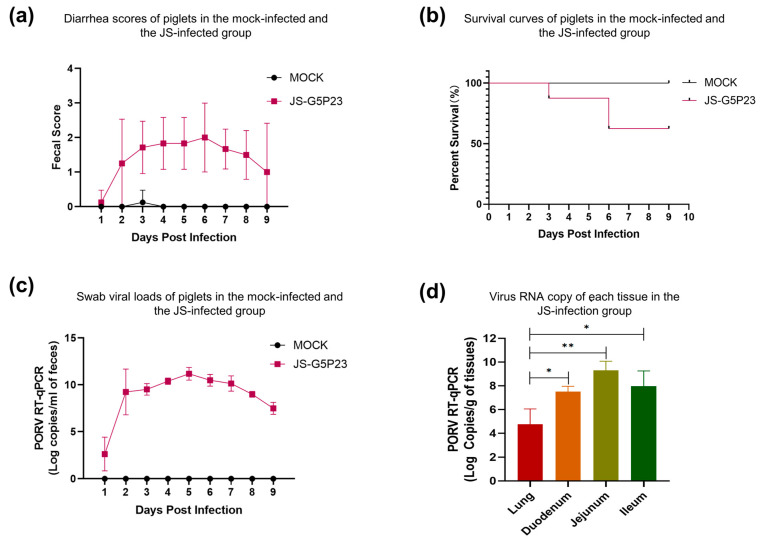
(**a**) Diarrhea scores of the piglets in the mock-infected and JS-infected groups. (**b**) Survival curves of the piglets in both groups. (**c**) Viral loads in the swabs of the piglets in both groups. (**d**) The viral RNA copy number per mg of each tissue collected from the piglets in both groups. Asterisk (*) indicates significant differences between different tissues. (* *p* < 0.05; ** *p* < 0.01).

## Data Availability

The data that support the findings of this study are available from the corresponding author upon reasonable request. GenBank accession numbers for the sample 14 RV sequences are OR644644–OR644654.

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
