# Peer review of "Isolation and Pathogenicity Analysis of a G5P[23] Porcine Rotavirus Strain"

_viruses, 2023, doi:10.3390/v16010021_

Round 1

Reviewer 1 Report

Comments and Suggestions for Authors

This is a well written report on the isolation and pathogenicity analysus of porcine rotavirus in China.

The manuscript has a very solid content. I have several suggestions which may improve the quality of the manuscript.

1. What is limitation of the study? No limitations of the study are indicated.

2. Is G5[P23] a novel strain? If not, please describe compare and analyze with the existing G5[P23] strain.

3. If possible, please describe the cut-off values for nt and aa identity between genotypes.

Author Response

Dear Reviewer:

Thank you very much for handling our manuscript for review. We have spent a lot of time carefully revising and supplementing your questions. Our point to point response the reviewers is list as following.

Reviewer: 1

This is a well written report on the isolation and pathogenicity analysus of porcine rotavirus in China.

The manuscript has a very solid content. I have several suggestions which may improve the quality of the manuscript.

  1. What is limitation of the study? No limitations of the study are indicated.

R:Thank you for your guidanceThe isolated strain in this study comes from Jiangsu Province, China and may not represent the porcine rotavirus in other regions or countries. The limitations of this study have been supplemented in the manuscript.

  1. Is G5[P23] a novel strain? If not, please describe compare and analyze with the existing G5[P23] strain.

R:To our knowledge, the G5P[23] strain was first discovered in piglets in 2013, but there is currently no research on the pathogenicity of the G5P[23] genotype rotavirus in piglets. We analyzed the homology between the current rotavirus isolate and the G5 and P[23] genes in the sequence analysis section, and compared the differences in pathogenicity of this isolate and the previously reported porcine rotavirus to piglets in the discussion section. We hope to receive your approval.

  1. If possible, please describe the cut-off values for nt and aa identity between genotypes.

R:In section 3.2 of the sequence analysis, the nucleotide identity cutoff values for rotavirus A genotypes were added, which were 80% (G), 80% (P), 85% (I), 83% (R), 84% (C), 81% (M), 79% (A), 85% (N), 85% (T), 85% (E), and 91% (H), respectively.

Reviewer 2 Report

Comments and Suggestions for Authors

Authors have isolated an RVA in which they claim that in NSP2 gene there is a recombination (Fig. 3) with that of human rotavirus.  Authors further characterizes this newly isolated RVA in 15-day-old piglets (which are kind of old for this study) regarding pathogenicity and histopathology.

This manuscript does not have a line number for references.  My general concerns focused on 2 aspects:

1.  the veterinary concerns.

2. the basic virology principle.

page 2, line 1 from top: what is Ga2+?  The most important mechanism for RVA to cause secretory diarrhea is that NSP4 serve as an enterotoxin that increase the chloride secretion through a calcium-dependent mechanism, and it also activate the enteric nervous system and block the intestinal sodium/glucose cotransporter.

page 3, first paragraph: after serial passage, the virus was not purified in any way (with limited-dilution or plaque cloning), therefore what you are studying is a heterogenous viral population, and when after genome sequencing, your claim that some genes are more homologous to human origin may be in doubt.

Section 2.8: you claim that these experimental pigs were tested negative for various viruses. It is good that you have thought of this, but this is a very serious statement, so you should explain how  you did that?  Also, you have to  indicate how the sows of these piglets were raised and vaccinated, and were these 15-old-piglets fed with colostrum which contain antibody? RVA usually infect neonates and using 15-day-old piglets is kind of  old.

section 3.1, last 2 lines: what is "bubble formation"? Is this intracellular vacuolation, cell swelling, or a large multinucleated cells after fusion (syncytium) or is this extracellular (i.e., an empty space after cell necrosis and detachment)?  What is loss of "motility" means?  Do you means cell stop growing further to fill the plastic surface?

page 5, line 3: please confirm that at which stage of life cycle, RVA can replicate in "nuclei" of cells.

page 6, section 3.2, paragraph 2: the sequenced virus was not purified.

section 3.4 (page 7 and 8), figure 4: authors need to consult an anatomic veterinary pathologist for the writing and interpretation of results.  For example, line 3, "swelling" of the lung. Normally lung has to swell in order to exchange air.  In pathology, when describing interstitial pneumonia, we use the term "rubbery texture" that means the lung is not collapsing after opening the  rib cage. For another example, "vacuolization of small intestinal cells".  This term is rather non-specific, because they are many types of cells in small intestine, including villi enterocyte, lymphocyte, lacteal duct, and fibroblast and perhaps smooth muscle cell in lamina propia.  Here vacuolation should occur in villi or villous enterocyte to be specific. for another example, "the number of goblet cells in the mucosa layer of the tissue was conspicuously reduced".  This may not be true because goblet cells although present in small intestine, they are more of a feature of large intestine. Further anatomic pathologist will use "mucosa" rather than "mucosal layer of the tissue", and more.

Figure 4: The distribution of RVA antigen in upper 2/3 of the villi is correct.  But here the villi lengths are shorter than expected even in mock treated pigs.  Further, in the lung, the RVA antigen is not impressive, although it is possible extraintestinal antigen is possible.

Author Response

Dear Reviewer:

Thank you very much for handling our manuscript for review. We have spent a lot of time carefully revising and supplementing your questions. Our point to point response the reviewers is list as following.

Reviewer: 2

Authors have isolated an RVA in which they claim that in NSP2 gene there is a recombination (Fig. 3) with that of human rotavirus.  Authors further characterizes this newly isolated RVA in 15-day-old piglets (which are kind of old for this study) regarding pathogenicity and histopathology.

This manuscript does not have a line number for references.  My general concerns focused on 2 aspects:

  1. the veterinary concerns.
  2. the basic virology principle.

1.page 2, line 1 from top: what is Ga2+?  The most important mechanism for RVA to cause secretory diarrhea is that NSP4 serve as an enterotoxin that increase the chloride secretion through a calcium-dependent mechanism, and it also activate the enteric nervous system and block the intestinal sodium/glucose cotransporter.

R:Thank you for your detailed explanation! I have made corresponding modifications in this section.

2.page 3, first paragraph: after serial passage, the virus was not purified in any way (with limited-dilution or plaque cloning), therefore what you are studying is a heterogenous viral population, and when after genome sequencing, your claim that some genes are more homologous to human origin may be in doubt.

R:Thank you for your reminder. We purified the virus through limited dilution and conducted multiple sequencing analyses on the virus, which are further explained on page 3.

3.Section 2.8: you claim that these experimental pigs were tested negative for various viruses. It is good that you have thought of this, but this is a very serious statement, so you should explain how  you did that?  Also, you have to  indicate how the sows of these piglets were raised and vaccinated, and were these 15-old-piglets fed with colostrum which contain antibody? RVA usually infect neonates and using 15-day-old piglets is kind of  old.

R:Thank you for your suggestion! We have provided additional explanations for this section. Firstly, we purchased piglets from a commercial pig farm without a history of PoRV outbreaks or vaccination. Secondly, we tested negative for pathogens and antibodies against Porcine Rotavirus, Porcine Epileptic Diarra virus, Transient Gastroenteris virus, Porcine Respiratory and Reproductive Syndrome virus, Porcine Circovirus, Classic Swine Fever virus, and Pseudorabies virus using Quantitative real-time polymerase chain reaction (qRT PCR) and Enzyme linked immunosorbent assay (ELISA). During the experiment, we used artificially raised milk instead of breastfeeding.

4.section 3.1, last 2 lines: what is "bubble formation"? Is this intracellular vacuolation, cell swelling, or a large multinucleated cells after fusion (syncytium) or is this extracellular (i.e., an empty space after cell necrosis and detachment)?  What is loss of "motility" means?  Do you means cell stop growing further to fill the plastic surface?

R:Thank you for pointing out our mistake! I did not describe this part accurately enough and made corresponding modifications. After being infected with the JS isolate, MA104 cells may exhibit vacuoles, stop growing, and shed cells.

5.page 5, line 3: please confirm that at which stage of life cycle, RVA can replicate in "nuclei" of cells.

R:Thank you for pointing out our mistake! After our discussion, the description here is inaccurate and has been deleted.

6.page 6, section 3.2, paragraph 2: the sequenced virus was not purified.

R:The virus purification method has been supplemented in Page 3

7.section 3.4 (page 7 and 8), figure 4: authors need to consult an anatomic veterinary pathologist for the writing and interpretation of results.  For example, line 3, "swelling" of the lung. Normally lung has to swell in order to exchange air.  In pathology, when describing interstitial pneumonia, we use the term "rubbery texture" that means the lung is not collapsing after opening the  rib cage. For another example, "vacuolization of small intestinal cells".  This term is rather non-specific, because they are many types of cells in small intestine, including villi enterocyte, lymphocyte, lacteal duct, and fibroblast and perhaps smooth muscle cell in lamina propia.  Here vacuolation should occur in villi or villous enterocyte to be specific. for another example, "the number of goblet cells in the mucosa layer of the tissue was conspicuously reduced".  This may not be true because goblet cells although present in small intestine, they are more of a feature of large intestine. Further anatomic pathologist will use "mucosa" rather than "mucosal layer of the tissue", and more.

R:Thank you for your careful guidance! I have provided a new description for this section. The HE results showed thickening of alveolar walls in piglets infected with JS (as shown by the green arrow in Figure 4b); The intestinal villi in the tissue mucosal layer are significantly shortened, and some mucosal epithelial cells can be seen to undergo pyknosis and necrosis (as shown by the yellow arrow in Figure 4b). The tissue shows obvious infiltration of inflammatory cells (as shown by the black arrow in Figure 4b)

8.Figure 4: The distribution of RVA antigen in upper 2/3 of the villi is correct.  But here the villi lengths are shorter than expected even in mock treated pigs.  Further, in the lung, the RVA antigen is not impressive, although it is possible extraintestinal antigen is possible.

R:Thank you for your question. The length of intestinal villi in piglets is shorter than expected, which may be due to piglets not adapting to artificially fed milk. In section 3.5, the antigen content in the lungs was compared with that in the small intestine. The lungs may not be the main target organ of porcine rotavirus, but during the autopsy process, we found that most piglets in the JS infection group had lesions in the lungs as shown in Figure 4a, but this situation did not occur in the simulated infection group. We had to think about this aspect.

Reviewer 3 Report

Comments and Suggestions for Authors

Porcine Rotaviruses (PoRVs) is the leading cause of severe intestinal diseases in piglets in Chinese farms and leads to significant economic losses. The authors of this study conducted clinical testing on a farm in Jiangsu Province, China, they collected fecal samples from 28 piglets, successfully isolated a G5P[23] genotype PoRV named RVA/Pig/China/JS/2023/G5P23. Genomic analysis revealed that the VP3, VP7, NSP2, and NSP4 genes of the JS strain were closely related to human RVAs, whereas the remaining gene segments were closely related to porcine RVAs. The JS strain inoculation resulted in a 100% diarrheal rate and a 37.5% mortality rate in 15-day-old piglets. These findings add to the understanding of the virulence of the G5P[23] genotype strain. and provided the information about current prevalence of rotavirus in pig in JiangSu, China.  However, there are some concerns need to be addressed.

1.     The authors claimed that all fecal samples tested positive for RVAs in the abstract, however, there is no specific information in the manuscript. It is recommended that the authors provide more details regarding the cases of rotavirus infection in the farm, including the positive rate of fecal samples. This additional information would enhance the transparency and credibility of the study.

2.     In section 3.1. it would be helpful for readers to understand the study if the authors can report the number of PoRV positive samples that were inoculated into MA 104 cells, and how many are those samples were positive in cell culture.

3.     The titles of Figure 5a and 5b appear incorrect, please revise them carefully.

4.     On page 6, the authors state that “The typical RV particles were observed in the cell culture medium (Fig.1c). The viral particles appeared wheel-shaped, with a diameter of 80–100 nm,”. However, the virus particles shown in Fig. 1c appear smaller than 80nm according to the scale bar. The author should explain how the measurement of 80 to 100nm were determined.

5.     It would be useful to include a discussion of the differences between animal challenges and clinical cases. Extending on these distinctions would improve readers' understanding of the study design and the relevance of the findings to real-world scenarios.

6.     The study is based on samples collected from a specific farm in Jiangsu Province, China. The findings may not be representative of PoRVs in other regions or countries. To strengthen the conclusion section, the authors should acknowledge the limitation that the findings may not be representative of PoRVs in other regions or countries. This acknowledgment will contribute to a more accurate interpretation of the results.

Comments on the Quality of English Language

The language is generally clear, but a few sentences could be made shorter. To improve readability, consider removing unnecessary words. 

Ensure that terminology is used consistently throughout the manuscript.

Author Response

Dear Reviewer:

Thank you very much for handling our manuscript for review. We have spent a lot of time carefully revising and supplementing your questions. Our point to point response the reviewers is list as following.

Reviewer: 3

Porcine Rotaviruses (PoRVs) is the leading cause of severe intestinal diseases in piglets in Chinese farms and leads to significant economic losses. The authors of this study conducted clinical testing on a farm in Jiangsu Province, China, they collected fecal samples from 28 piglets, successfully isolated a G5P[23] genotype PoRV named RVA/Pig/China/JS/2023/G5P23. Genomic analysis revealed that the VP3, VP7, NSP2, and NSP4 genes of the JS strain were closely related to human RVAs, whereas the remaining gene segments were closely related to porcine RVAs. The JS strain inoculation resulted in a 100% diarrheal rate and a 37.5% mortality rate in 15-day-old piglets. These findings add to the understanding of the virulence of the G5P[23] genotype strain. and provided the information about current prevalence of rotavirus in pig in JiangSu, China.  However, there are some concerns need to be addressed.

  1. The authors claimed that all fecal samples tested positive for RVAs in the abstract, however, there is no specific information in the manuscript. It is recommended that the authors provide more details regarding the cases of rotavirus infection in the farm, including the positive rate of fecal samples. This additional information would enhance the transparency and credibility of the study.

R:The question you raised is very valuable, but we deeply regret it. All the diarrhea samples we collected this time were positive for porcine rotavirus, and we provided a brief description in the manuscript. However, we only collected a small sample size, which cannot accurately represent the actual prevalence of porcine rotavirus in pig farms and the region. This article mainly studies the isolation of rotavirus and its pathogenicity to piglets.

  1. In section 3.1. it would be helpful for readers to understand the study if the authors can report the number of PoRV positive samples that were inoculated into MA 104 cells, and how many are those samples were positive in cell culture.

R:We have provided additional explanations in section 3.1. We processed all 28 positive samples and inoculated them into MA104 cells, with only one sample able to stably passage in MA104 cells.

  1. The titles of Figure 5a and 5b appear incorrect, please revise them carefully.

R:Thank you for your careful reminder. We have made the corresponding modifications.

  1. On page 6, the authors state that “The typical RV particles were observed in the cell culture medium (Fig.1c). The viral particles appeared wheel-shaped, with a diameter of 80–100 nm,”. However, the virus particles shown in Fig. 1c appear smaller than 80nm according to the scale bar. The author should explain how the measurement of 80 to 100nm were determined.

R:We have replaced the images with clearer ones and provided new descriptions, hoping to receive your approval.

  1. It would be useful to include a discussion of the differences between animal challenges and clinical cases. Extending on these distinctions would improve readers' understanding of the study design and the relevance of the findings to real-world scenarios.

R:Thank you for your suggestion. We have added a description of this aspect in the discussion section.

6.The study is based on samples collected from a specific farm in Jiangsu Province, China. The findings may not be representative of PoRVs in other regions or countries. To strengthen the conclusion section, the authors should acknowledge the limitation that the findings may not be representative of PoRVs in other regions or countries. This acknowledgment will contribute to a more accurate interpretation of the results.

R:Thank you for your reminder. I have provided additional explanations in section 4.

Reviewer 4 Report

Comments and Suggestions for Authors

comments are along the sentences of the article (unfortunately there is numeration)

page 2: The outer shell proteins VP7 and VP4 induce neutralizing antibodies: the reference is strictly related to human RoV. It is suggested to add a reference more specific (SN-Ab in vaccinated pigs, for example)

page 2; page 4, point 2.8:  used to infect 15-day-old: why 2 weeks old piglets instead of most common "target" for Rov: 1st week old piglets ?

page 4, point 2.8: Ethics Committee approval: is there a statistic evaluation/consideration about the 3 R principles ? why 8+8 piglets were necessary for the test ? 

page 7; point 3.4: how much of the 8-control piglets were sacrified ?

page 11, Conclusion: Ethics approval ..... Authors should compare if the standards in the "Guide for the Care and Use of Laboratory Animals of the Ministry of Science and Technology of the People’s Republic of China" may be consifdered in line with EU/USA standards; in particular, the issue of numerosity of infected, sacrified piglets should be better clarified. 

Author Response

Dear Reviewer:

Thank you very much for handling our manuscript for review. We have spent a lot of time carefully revising and supplementing your questions. Our point to point response the reviewers is list as following.

Reviewer: 4

comments are along the sentences of the article (unfortunately there is numeration)

1.page 2: The outer shell proteins VP7 and VP4 induce neutralizing antibodies: the reference is strictly related to human RoV. It is suggested to add a reference more specific (SN-Ab in vaccinated pigs, for example)

R: Thank you for your guidance! I have replaced it with literature related to porcine rotavirus.

  1. page 4, point 2.8:  used to infect 15-day-old: why 2 weeks old piglets instead of most common "target" for Rov: 1st week old piglets ?

R:Because we found that two week old piglets in infected pig farms are also infected with porcine rotavirus, we want to explore the pathogenicity of porcine rotavirus on 2-week-old piglets.

3.page 4, point 2.8: Ethics Committee approval: is there a statistic evaluation/consideration about the 3 R principles ? why 8+8 piglets were necessary for the test ? 

R:Because not all piglets infected with porcine rotavirus will get sick in the investigation of porcine rotavirus, the number of samples is too low to accurately study the incidence rate and mortality.

  1. 4:page 7; point 3.4: how much of the 8-control piglets were sacrified ?

R:I made a supplement in section 2.8. Three piglets from the virus-infected group and three piglet from the mock group were euthanized at 9 d postinfection (dpi).

  1. page 11, Conclusion: Ethics approval ..... Authors should compare if the standards in the "Guide for the Care and Use of Laboratory Animals of the Ministry of Science and Technology of the People’s Republic of China" may be consifdered in line with EU/USA standards; in particular, the issue of numerosity of infected, sacrified piglets should be better clarified. 

R: Thank you for your reminder. I have made relevant supplements in section 2.8.

Round 2

Reviewer 2 Report

Comments and Suggestions for Authors

The revised version of the manuscript have improved.

Section 2.5: convert all "rpm" into "g" force.

Section 2.8, page 4 line 5 from bottom: for histopathology the tissues are routinely fixed in 4% "formaldehyde".

Section 3.1, sentence 1: after testing of ??? sample with ??? assay.

page 9, paragraph 1:

- "The lung showed thickening of the alveolar walls suggestive of an interstitial pneumonia"

-"The intestinal villi are significantly shortened and blunted indicative of villous atrophy, and the villous enterocytes can be seen to undergo necrosis".  Villi are normally present in intestinal mucosa, you do not need to describe it.

- " the villi lamina propia shows an increased number of inflammatory cells predominantly lymphocytes".  In normal condition, the lamina propia has a number of surveillance lymphocytes for diet antigens, so here what you see is "an increased number".

page 10, bottom: modified your histopathology description as above.

reference 9: 105-111.

reference 17: 491-495.

reference 20: BMC.

reference 38: British medical journal or BMJ.

Author Response

Dear Reviewer:

Thank you for carefully reviewing our manuscript again and providing valuable revision suggestions.We have made revisions to the manuscript according to your suggestions.Our point to point response the reviewers is list as following.

Comments and Suggestions for Authors:

The revised version of the manuscript have improved.

  1. Section 2.5: convert all "rpm" into "g" force.

R:We converted "rpm" to "g" in the manuscript.

  1. Section 2.8, page 4 line 5 from bottom: for histopathology the tissues are routinely fixed in 4% "formaldehyde".

R:We have modified "paraformaldehide" to "formaldehide" in the manuscript.

  1. Section 3.1, sentence 1: after testing of ??? sample with ??? assay.

R:Thank you for your careful reminder. We have provided a new description of this section in the manuscript.

  1. page 9, paragraph 1:

- "The lung showed thickening of the alveolar walls suggestive of an interstitial pneumonia"

-"The intestinal villi are significantly shortened and blunted indicative of villous atrophy, and the villous enterocytes can be seen to undergo necrosis".  Villi are normally present in intestinal mucosa, you do not need to describe it.

- " the villi lamina propia shows an increased number of inflammatory cells predominantly lymphocytes".  In normal condition, the lamina propia has a number of surveillance lymphocytes for diet antigens, so here what you see is "an increased number".

R:Thank you for your guidance. We have made modifications to this section according to your description.

  1. page 10, bottom: modified your histopathology description as above.

R: A new description has been made for this section as mentioned above.

  1. reference 9: 105-111.

reference 17: 491-495.

reference 20: BMC.

reference 38: British medical journal or BMJ.

R: Thank you for pointing out our mistake carefully. We have made revisions in the manuscript.

Thank you again for your guidance on our above mistakes!
